# Electron donation of non-oxide supports boosts O$_2$ activation on nano-platinum catalysts

Tao Gan[1], Jingxiu Yang [2✉], David Morris [3], Xuefeng Chu[4], Peng Zhang [3], Wenxiang Zhang[1✉], Yongcun Zou[1], Wenfu Yan [1], Su-Huai Wei [5] & Gang Liu [1✉]

Activation of O$_2$ is a critical step in heterogeneous catalytic oxidation. Here, the concept of increased electron donors induced by nitrogen vacancy is adopted to propose an efficient strategy to develop highly active and stable catalysts for molecular O$_2$ activation. Carbon nitride with nitrogen vacancies is prepared to serve as a support as well as electron sink to construct a synergistic catalyst with Pt nanoparticles. Extensive characterizations combined with the first-principles calculations reveal that nitrogen vacancies with excess electrons could effectively stabilize metallic Pt nanoparticles by strong p-d coupling. The Pt atoms and the dangling carbon atoms surround the vacancy can synergistically donate electrons to the antibonding orbital of the adsorbed O$_2$. This synergistic catalyst shows great enhancement of catalytic performance and durability in toluene oxidation. The introduction of electron-rich non-oxide substrate is an innovative strategy to develop active Pt-based oxidation catalysts, which could be conceivably extended to a variety of metal-based catalysts for catalytic oxidation.

[1] State Key Laboratory of Inorganic Synthesis and Preparative Chemistry, College of Chemistry, Electron Microscopy Center, Jilin University, Changchun, China. [2] Key Laboratory for Comprehensive Energy Saving of Cold Regions Architecture of Ministry of Education, School of Materials Science and Engineering, Jilin Jianzhu University, Changchun, China. [3] Department of Chemistry, Dalhousie University, Halifax, Canada. [4] Jilin Provincial Key Laboratory of Architectural Electricity & Comprehensive Energy Saving, School of Electrical and Electronic Information Engineering, Jilin Jianzhu University, Changchun, China. [5] Beijing Computational Science Research Center, Beijing, China. ✉email: yangjingxiu@jlju.edu.cn; zhwenx@jlu.edu.cn; lgang@jlu.edu.cn

Catalytic oxidation encompasses a great deal of important processes in the chemical industry and emissions control, such as ethylene epoxidation, CO, and hydrocarbon conversion[1–5]. These processes account for a considerable amount of global chemical production, especially the energy consumption reactions[2]. Platinum is a widely used metal reactive centers of catalysts for emissions control[4–6]. Considering the cost of the catalysts and energy consumption of the reaction, one desires to achieve complete conversion of pollutants at the lowest possible reaction temperature with lowest amount of Pt loading[7–9].

Activation of molecular oxygen is a critical step for heterogeneous catalytic oxidation[2,10–12]. The reactivity of $O_2$ adsorption and dissociation (two successive steps in $O_2$ activation) on the catalyst surface relies on the charge transfer from the metal surface, and/or metal-support interface to the stable $O_2$ molecules[2,10]. Sufficient charge transfer would weaken the O–O bond and render it more reactive toward reactant molecules[2]. Therefore, introducing excess electrons to the Pt centers and/or the Pt-support interface should be a practical strategy to obtain a highly active Pt-based catalyst.

Oxides materials are widely adopted as supports to stabilize the Pt nanoparticles[13–18]. Surface oxygen vacancies could serve as electron donation sites to enhance the performance of $O_2$ activation[19]. However, the metallicity of Pt nanoparticles inevitably decrease during the loading process onto the oxides support[20]. The solid-state reaction between Pt nanoparticles and the oxide support cause the oxidation of the neutral Pt atoms to cationic atoms and this, in turn, can weaken the charge transfer for $O_2$ activation[8].

As a non-oxide and metal-free material, carbon nitride (especially g-$C_3N_4$) has become a star material in areas of photocatalysis and electrocatalysis in the past decade[21–26]. Basically, carbon nitride is a class of polymeric materials consisting mainly of carbon and nitrogen. The conjugated polymeric network endows carbon nitride with desirable physicochemical properties, such as insolubility in acidic, neutral, or basic solvents[26]. In comparison with conventional carbon materials, inertness against oxidation is an important feature of carbon nitride[22]. However, this attractive property negatively affects their performance in heterogeneous catalysis. The inert surface renders it less reactive toward reactants and few active heterogeneous oxidation catalysts have been fabricated based on carbon nitride materials.

In this work, the potential advantage of carbon nitride was fully amplified through manipulating the nitrogen-vacancy of carbon nitride. The increased electron donors induced by nitrogen vacancy was demonstrated as an efficient strategy to develop highly active and stable Pt-based catalysts for molecular $O_2$ activation. The complementary characterization techniques and the first-principles calculations reveal that nitrogen vacancies not only possesses extraordinary capability to stabilize the metallic Pt nanoparticles, but also assists to donate electrons to the adsorbed $O_2$. In catalytic toluene oxidation, the catalyst with Pt loading as low as 0.3 wt% exhibits excellent catalytic performance, which could completely convert toluene (1000 ppm) to $CO_2$ at about 190 °C ($T_{100}$). This conversion temperature is about 30 °C lower than that of the well-known Pt/$CeO_2$ with the same amount of Pt loading, and the catalyst effectively reduce the activation energy of this reaction. In addition, the catalyst exhibits water-resistant properties and unexpected stability. Kept in air for 1 year as well as after hydrothermal aging treatment, the catalyst maintained the high activity. This work demonstrated an alternative concept to develop highly active and stable Pt-based oxidation catalysts. The prototype non-oxide support exemplified here could be extended to prepare other efficient metal-based catalysts.

## Results

**Identifying the formation of nitrogen vacancies.** Nitrogen vacancies could be engineered by controlling the polymerization degrees of carbon nitride. We selected synthetic conditions that favor the higher condensation of melamine into corresponding products (temperatures between 500 and 650 °C), aiming to meet the stability requirement of application in vapor-phase reactions. The resultant samples are denoted as CN500, CN525, CN600, and CN650 according to the preparation temperatures. Two possible vacancy sites including vacancy of $N_{2c}$ and $N_{3c}$ are shown and denoted as $V_{N2c}$ and $V_{N3c}$, respectively, in Fig. 1a. According to our thermodynamic calculation, the formation of the neutral defects $V_{N2c}$ and $V_{N3c}$ are equally possible due to the slight energy difference of 0.025 eV/site. According to the bader charge analysis, the extra electrons introduced by $V_{N2c}$ or $V_{N3c}$ are not localized on the dangling carbon atoms, but re-distributed on the adjacent carbon atoms via the π-conjugated networks of polymeric carbon nitride.

The XRD patterns show that CN500 is an oligomer sample containing a melem related structure (Fig. 1b)[27–30]. The samples of CN525, CN600, and CN650 possess typical melon structure[27,31,32]. The structure change is due to the condensation of framework at higher temperature. That directly influence interplanar stacking distance, long-range order of the in-plane structural packing, specific surface area and surface defects (Supplementary Table 1, Supplementary Figs. 1 and 2).

X-ray photoelectron spectroscopy (XPS) was carried out to detect the atomic structure changes on the surface of these samples. The C 1s peak at 288.1 eV can be assigned to a $C(N)_3$ coordination[31–33], accompanied with a signal from adventitious carbon at 284.8 eV (Fig. 1c). The sharp N 1s XPS signal with a shoulder (Fig. 1d) can be fitted into peaks assigned to C–N–H (401.0 eV), N-$(C)_3$ (400.0 eV), and C–N=C (398.8 eV) (Supplementary Table 2)[32,34–36]. Both the shift of N 1s XPS signal and changes of the ratio between C–N=C (denoted as $N_{2C}$, see in Fig. 1a) and N-$(C)_3$ (denoted as $N_{3C}$, see in Fig. 1a) can be observed from the spectra. The $N_{2C}/N_{3C}$ ratio drops drastically from 8.88 of CN500 to 3.42 of CN650 (Fig. 1e). Correspondingly, the surface C/N atomic ratio increase from 0.65 to 0.73 (Supplementary Table 3). These experimental results indicate that surface nitrogen vacancies are formed during the condensation at high temperature and are mainly located at the $N_{2c}$ lattice sites. In agreement with the theoretical expectation, extra electrons are found on some of the carbon atoms, which is further confirmed by the shift of C 1s peak to the low binding energy (Fig. 1c).

The optical characters further verify the formation of nitrogen vacancies in the samples prepared at high temperature. The color changes from pale yellow (CN500, Fig. 1f) to light brown (CN650). In agreement with the observed color, a gradual red-shifted absorption threshold can be observed in the samples of CN525, CN600, and CN650 (Fig. 1g). This red-shifted absorption threshold is originated from the vacancy-induced delocalization of the π-electron in the conjugated system[37]. The gradual enhancement of this signal indicates the larger amount of nitrogen vacancies can be created at a higher preparation temperature. The photoluminescence (PL) spectra (Fig. 1h and Supplementary Fig. 3) show that CN500 exhibits a strong emission peak at 458 nm. This emission is mainly originated from the band-to-band recombination electrons and holes. With the increasing preparation temperatures, the intensity of the emission peak obviously decreases. The time-resolved PL spectra (Supplementary Fig. 4 and Supplementary Table 4) show that the lifetimes of charges in these samples also exhibit a decrease trend. Combined these results, it is obtained that nonradiative recombination increases in the samples prepared at high

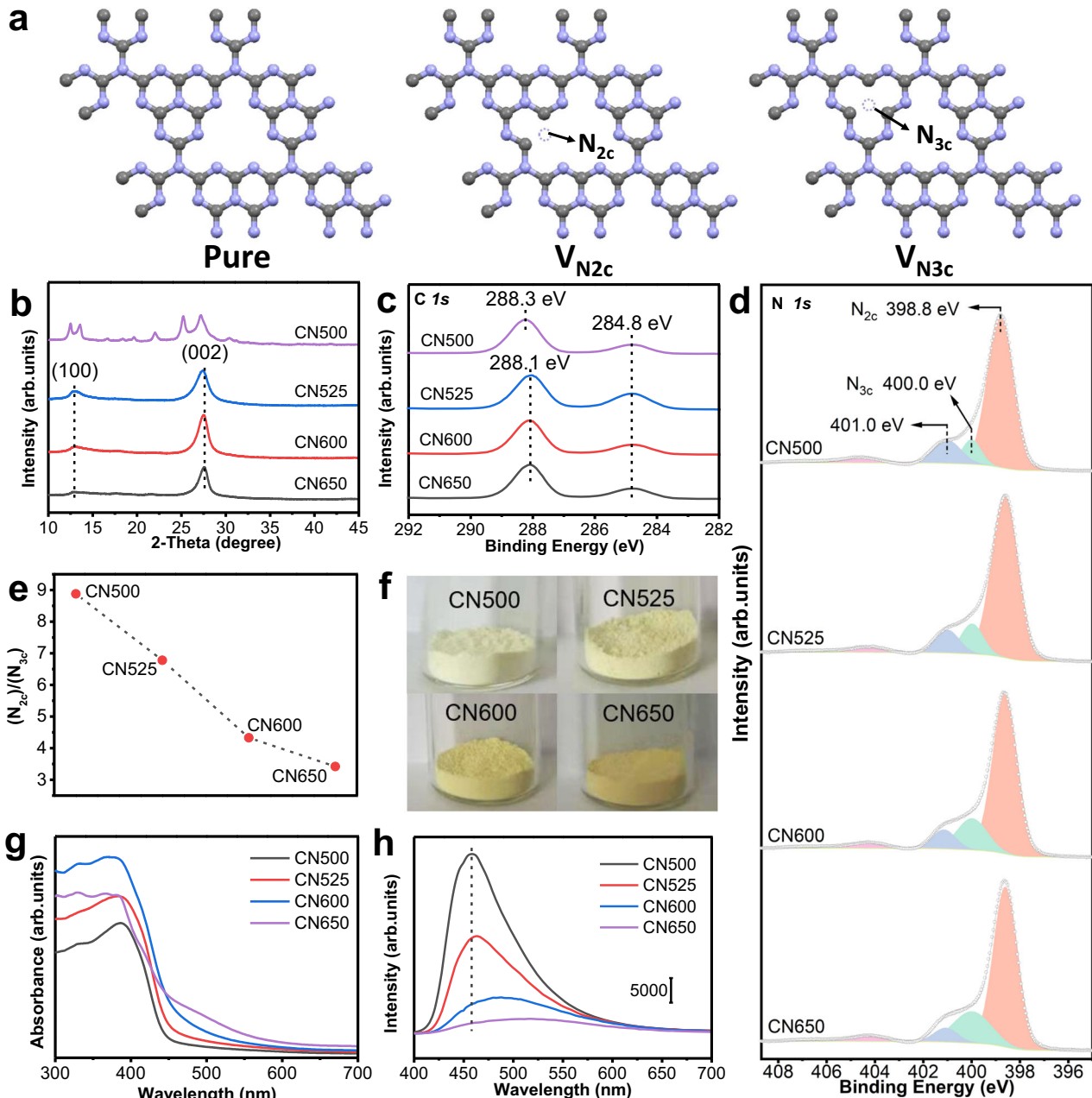

**Fig. 1 Structure, surface, and optical properties of different carbon nitride supports. a** Schematic structures of the carbon nitrides with and without nitrogen vacancies. (gray represents C, purple represents N). **b–h** Various characterizations: **b** X-ray diffraction (XRD) patterns. **c** C 1s X-ray photoelectron spectra (XPS). **d** N 1s XPS spectra. **e** The $(N_{2c})/(N_{3c})$ ratio calculated by the high-resolution N 1s XPS spectra. **f** Photographs of different samples. **g** UV–Vis diffuse reflectance spectra. **h** Photoluminescence (PL) emission spectra, excitation wavelength: 380 nm.

temperatures, and the formation of nitrogen vacancies occurs during the condensation at high temperature.

**Anchoring Pt nanoparticles on carbon nitride supports**. Pt nanoparticles prepared by a polyol reduction method were used as a precursor to synthesize the catalysts[8,38]. The size of Pt nanoparticles were 3–4 nm in the colloid (Supplementary Fig. 5). In typical catalysts, the loading amount of Pt was controlled at 0.3 wt%. These Pt nanoparticles exhibit different dispersion behaviors over the carbon nitrides prepared at different temperatures (Fig. 2). Both isolated and aggregated behaviors of Pt nanoparticles can be observed over CN500 support (Fig. 2a, b). No Pt diffraction signals can be observed in the XRD patterns (Supplementary Fig. 6), indicating that the aggregated Pt

nanoparticles maintain isolated and has not grown into large particles. As for the other three samples (CN525, CN600, and CN650), only isolated Pt nanoparticles (Fig. 2, marked by arrow, and Supplementary Fig. 7) can be observed. The statistical results show that these nanoparticles preserve the size of Pt nanoparticles in the colloid. The lattice fringe with a $d$ spacing of 0.23 nm was observed in the HRTEM images, which could be ascribed to the (111) plane of Pt nanoparticles. During the loading process, the structural and surface properties of carbon nitrides are almost not affected, which are ascertained by various measurements including XRD, FT-IR, SEM, XPS, UV–Vis, and PL spectra (Supplementary Figs. 6, 8–16, Supplementary Table 5 and 6).

Given the above characterization results, it can be concluded that the presence of nitrogen vacancies facilitates the dispersion of

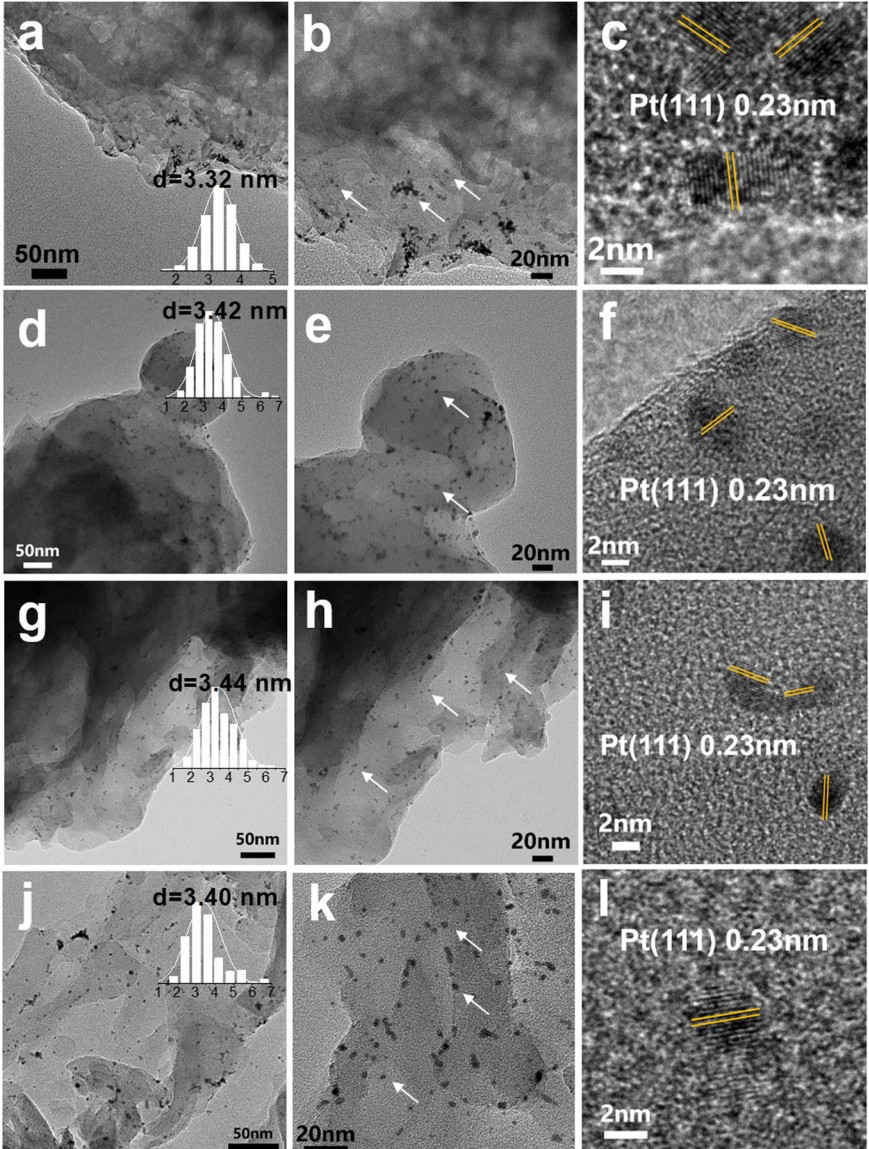

**Fig. 2 TEM and HRTEM images of different supported Pt samples. a–c** Pt/CN500. **d–f** Pt/CN525. **g–i** Pt/CN600. **j–l** Pt/CN650.

Pt on the surface of carbon nitride. This deduction could be further confirmed by our calculation result. According to the first-principles calculation, Pt atom can easily passivate the left dangling carbon atoms around $V_{N2c}$ by gaining −2.33 eV rather than $V_{N3c}$ by costing 0.43 eV. This result represents that $V_{N2c}$ should be responsible for anchoring Pt. This distinguished binding energy might be ascribed to the large space around $V_{N2c}$, which allows the p-d coupling of the Pt and the neighbored C and N atoms. More interestingly, merely slight charge transfer (about 0.23 e) is found between the Pt atom and the bonded atoms, which suggests the passivated Pt is more like a metal. The chemical states of Pt nanoparticles were analyzed using Pt 4*f* XPS spectra (Fig. 3a). All these four samples exhibit similar Pt 4*f* peaks, centering at 70.6 and 73.9 eV. Comparing with the oxides-supported-Pt catalysts, the binding energy obviously shift to a low value, indicating that most of Pt nanoparticles maintain the metallic state[13,16,18]. Synchrotron X-ray absorption experiments were also performed to further investigate the oxidation states of the Pt nanoparticles. In the X-ray absorption near edge structure (XANES) region, the edge energy of the samples of Pt/CN500 and Pt/CN650 are similar to that of Pt foil (Fig. 3b), indicating the

metallic character. The extended X-ray absorption fine structure (EXAFS) spectra of these samples in R-space were also compared (Fig. 3c). The EXAFS spectra of both samples contain a major peak corresponding to Pt–Pt, present at approximately 2.5 Å. The EXAFS spectra lacked a major peak representing a Pt–O bond, which would be located around 1.5 Å. This qualitative analysis helps to confirm the metallic state of Pt nanoparticles.

**Catalytic toluene oxidation over platinum–carbon nitride catalysts.** Catalytic toluene oxidation was carried out to detect the catalytic performance of these catalysts (Fig. 4). In absence of catalysts, no toluene conversion is observed in the test temperature region. Figure 4a shows that the toluene catalytic conversions over Pt/CN525, Pt/CN600, and Pt/CN650 exhibit S-shaped curves. The toluene conversion increases upon the increase of reaction temperature. Among these three samples, Pt/CN650 exhibits the highest activity and nearly all toluene is selectively converted to $CO_2$. $T_{50}$ (the reaction temperature of toluene with 50% conversion) and $T_{100}$ (complete conversion temperature of toluene) were achieved at 173 °C, 190 °C for Pt/CN650. The

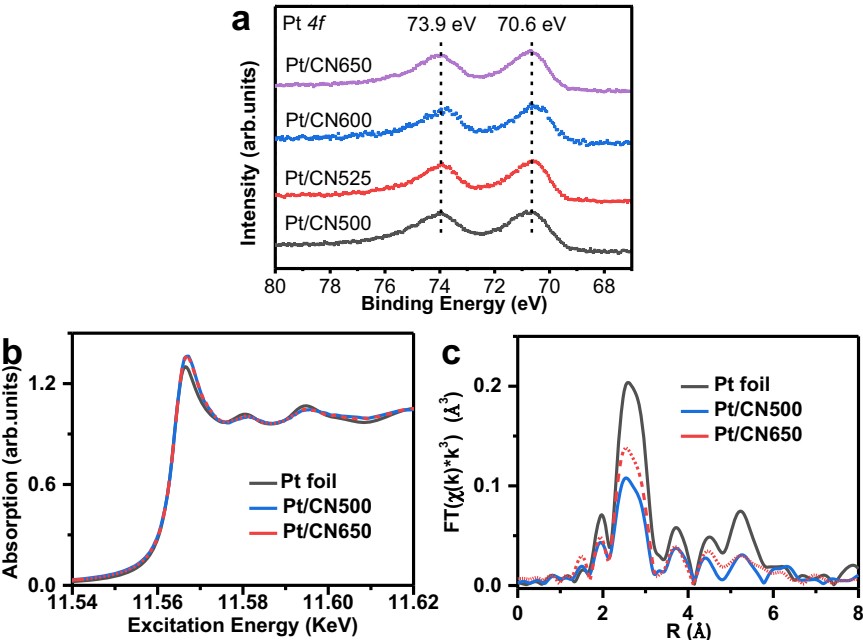

**Fig. 3 Chemical states of Pt nanoparticles over different carbon nitride supports. a** Pt 4f XPS spectra. **b, c** X-ray absorption spectra. **b** X-ray absorption near edge structure (XANES) region. **c** The $k^3$-weighted FT-EXAFS results.

reaction temperature is about 30 °C lower than that of Pt/CeO$_2$ with the same amount of Pt loading (Fig. 4a). CeO$_2$ is a well-known redox metal-oxide support and has been recognized as a key component in the three-way catalytic converters for O$_2$ activation[39,40]. In our case, CN650 obviously has advantages over CeO$_2$. Besides, the performance of Pt/CN650 is even comparable with the samples with high Pt-loading (Supplementary Table 7).

It should be noted that Pt/CN650 also exhibits high stability in the reaction process. Even after storage in air for 1 year, there is almost no change for the conversion of toluene in the test period at 190 °C (Fig. 4b). Moreover, water gas in the reactants has no effect on the catalytic performance of Pt/CN650 (Fig. 4c). This is a desirable property for the catalysts used in emissions control since water vapor inevitably exists in exhaust emission. It was reported that many catalysts would deactivate in a certain degree in the presence of water gas[4]. Besides, we have also tested the catalytic performance of Pt/CN650 after hydrothermal aging treatment. There is almost no difference between fresh and treated samples (Supplementary Fig. 17). It should be noted that all above catalytic results are obtained at the low-Pt-contents of 0.3 wt%. Considering the cost of the catalysts and the energy consuming of the reaction, it takes a great advantage to achieve the complete conversion of toluene at such a low temperature and low amount of Pt loading.

The stability of catalytic active sites in the reaction process is a critical factor for practical application. For this catalyst, it includes the changes of the morphology and the chemical states of the Pt nanoparticles and the carbon nitride support. In this work, we systematically compared the properties of fresh Pt/CN650 and the one used for 20 h reaction (denoted as Pt/CN650-20h). The SEM, TEM images, and XRD patterns show that Pt/CN650-20h maintains the morphology and structure of Pt/CN650 (Fig. 5a, Supplementary Figs. 18, and 19). Pt nanoparticles are highly dispersed on the surface of the carbon nitride support and no aggregation is observed in the detect regions (Fig. 5a). As shown in the Pt 4f XPS, Pt maintains in metallic states after long-time reaction (Fig. 5b), which is ascertained by the XANES (Fig. 5d) and EXAFS measurements (Fig. 5e). More importantly, N 1s and C 1s results also show the well-maintained surface chemical states

of carbon nitride support (Fig. 5c and Supplementary Fig. 20). The ratio between C–N–H, N-(C)$_3$, and C–N=C is almost the same before and after the reaction (Supplementary Table 8). All above results show that the nitrogen vacancies and metallic Pt are both stable during the catalytic toluene oxidation.

Considering the effect of loading amount, we further investigated the catalytic oxidation of toluene over the catalysts with higher Pt loading (0.8 wt%). Under the same reaction conditions as 0.3 wt% Pt/CN650 (space velocity, SV = 24,000 mL g$^{-1}$ h$^{-1}$), $T_{50}$ and $T_{100}$ can be achieved at 166 °C, 180 °C over 0.8 wt% Pt/CN650 (Fig. 4d). Interestingly, these Pt/CN650 samples also exhibit high activity under high SV (60,000 mL g$^{-1}$ h$^{-1}$). Toluene could be completely converted to CO$_2$ at about 190 °C over 0.8 wt% Pt/CN650 under such high SV (Fig. 4e). The apparent activation energies are estimated by using the Arrhenius relationship (Fig. 4f). The activation energy over Pt/CN650 with different Pt contents is 46.2 kJ mol$^{-1}$ (0.8 wt%Pt/CN650), 46.3 kJ mol$^{-1}$ (0.3 wt%Pt/CN650) and 46.9 kJ mol$^{-1}$ (0.1 wt%Pt/CN650), respectively. To our knowledge, this value is much lower than that of the most reported oxides-supported catalysts including the optimized Pt/Al$_2$O$_3$ catalyst recently reported by our group[8,41–44]. Besides, these Pt/CN catalysts show very similar apparent activation energy, which indicates that the increase of Pt loading amount mainly improves the numbers of active sites for catalytic oxidation.

In comparison, over 0.3 wt% Pt/CN500, only 12.3% toluene conversion can be observed when the reaction temperature is up to 220 °C (Fig. 4a). Increasing the amount of Pt to 0.8 wt% over CN500 has no significant impact on the catalytic activity. (Fig. 4d). One possible reason is the aggregation of Pt nanoparticles over CN500 support. However, it is insufficient to explain the low activity of Pt/CN500 since a certain amount of isolated Pt nanoparticles can be observed on the surface of CN500 (Fig. 2b). The morphology of the Pt is similar to that of the sample Pt/CN650 (Fig. 2k). Both the XPS and the X-ray absorption results show that Pt nanoparticles possess similar metallic properties in the samples of Pt/CN500 and Pt/CN650 (Fig. 3 and Supplementary Fig. 21). We also detected the catalytic performance of CO oxidation on these catalysts. It exhibits

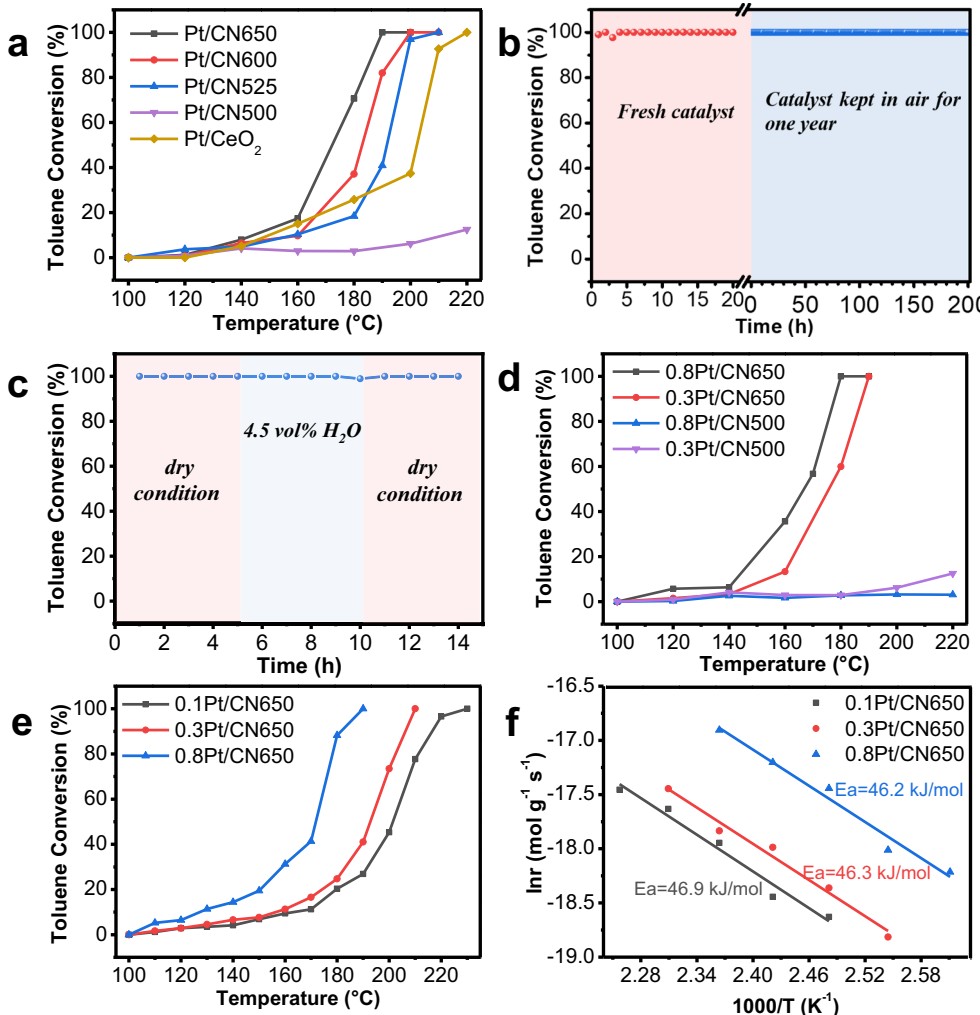

**Fig. 4 Catalytic activities of toluene oxidation over different supported Pt catalysts. a** Toluene conversion as a function of temperature, Pt loading amount 0.3 wt%, reaction conditions: $C_7H_8$ 1000 ppm, space velocity (SV) = 24,000 mL $g^{-1}$ $h^{-1}$, air balance. **b** Stability test of Pt/CN650 for toluene oxidation, reaction temperature: 190 °C, the other reaction conditions are the same as **a**. **c** Influence of gas-phase water on toluene oxidation over Pt/CN650, $H_2O$ 4.5 vol%, the other reaction conditions are the same as **b**. **d** Toluene conversion over Pt/CN650 and Pt/CN500 with different loading amount of Pt, reaction conditions are the same as **a**. **e** Toluene conversion as a function of temperature over Pt/CN650 with different loading amount of Pt at SV of 60,000 mL $g^{-1}$ $h^{-1}$, the other reaction conditions are the same as **a**. **f** Arrhenius plots for toluene oxidation over Pt/CN650 with different loading amount of Pt.

similar activity trend to that in toluene oxidation (Supplementary Fig. 22), which is 0.8 wt%Pt/CN650 > 0.3 wt%Pt/CN650 » 0.8 wt% Pt/CN500 ≈ 0.3 wt%Pt/CN500. Undoubtedly, the metallic Pt nanoparticles serve as the active sites for CO adsorption. The different catalytic activities over different carbon nitride supports further implies the participation of the nitrogen vacancies in the reaction process.

Diffuse-reflectance infrared Fourier transform (DRIFT) analysis of toluene adsorbed onto the catalyst surface, and temperature-programmed desorption of toluene ($C_7H_8$-TPD) were carried out to investigate the performance of catalysts in the activation/adsorption of toluene (Supplementary Figs. 23 and 24). DRIFT spectra of toluene adsorption show that no obvious difference between two typical samples, Pt/CN500 and Pt/CN650 (Supplementary Fig. 23). In the profiles of $C_7H_8$-TPD, the $C_7H_8$ desorption area over Pt/CN500 is larger than that of Pt/CN650 (Supplementary Fig. 24). According to these two results, it can be ruled out that Pt/CN650 has an advantage on the activation of toluene in comparison with Pt/CN500. The influence of PVP could also be excluded. Similar reaction results could be obtained

in the absence of PVP during preparation (Supplementary Fig. 25). Considering the defect-state of carbon nitride supports, the participation of nitrogen vacancies in molecular $O_2$ activation should be responsible for the catalytic activities.

The adsorption and activation of $O_2$ on the catalyst is recognized as the key step of the whole reaction. To understand the mechanism of the enhanced $O_2$ activation in this system, we have performed the first-principles calculation of the $O_2$ adsorption on $C_3N_4(001)$, Pt-adsorbed $C_3N_4$ (001) ($C_3N_4(001)$ +Pt), Pt (111), and Pt passivated nitrogen vacancy ($V_{N2c}$ + Pt), respectively, as shown in Fig. 6a. The O–O bond length of the adsorbed $O_2$ molecule is 1.260, 1.307, 1.365, and 1.496 Å, respectively, on different substrates. The gradual elongated bond-lengths indicate the gradual enhanced activation of $O_2$ due to the electron donation of the substrates. Particularly when the $O_2$ is adsorbed on $V_{N2c}$ + Pt, it is found that about 0.65 electron transferred from the passivated Pt and the dangling carbon atoms to the $O_2$ molecule, which occupies the antibonding state ($\sigma_p*$) as shown in Fig. 6b top. Adsorbed solely on the Pt (111), Pt-adsorbed $C_3N_4$ (001), or $C_3N_4$ (001) surface, the charge

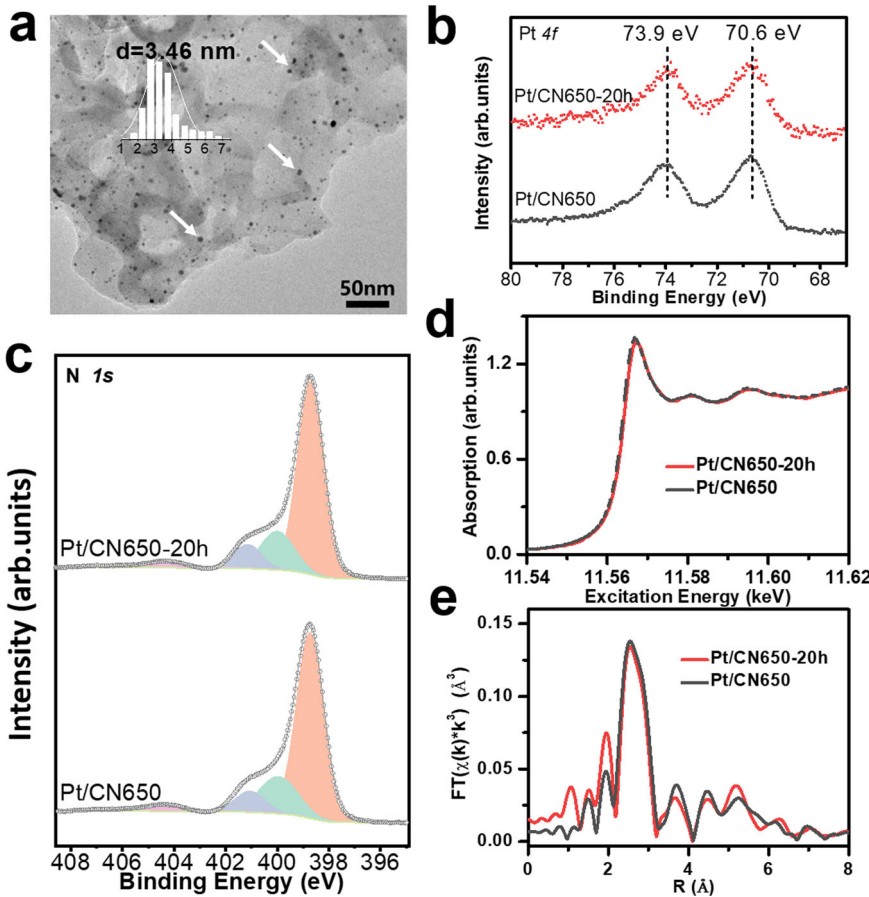

**Fig. 5 Structural and surface properties of fresh and used Pt/CN650. a** TEM image. **b** Pt *4f* XPS spectra. **c** N 1*s* XPS spectra. **d** XANES. **e** EXAFS.

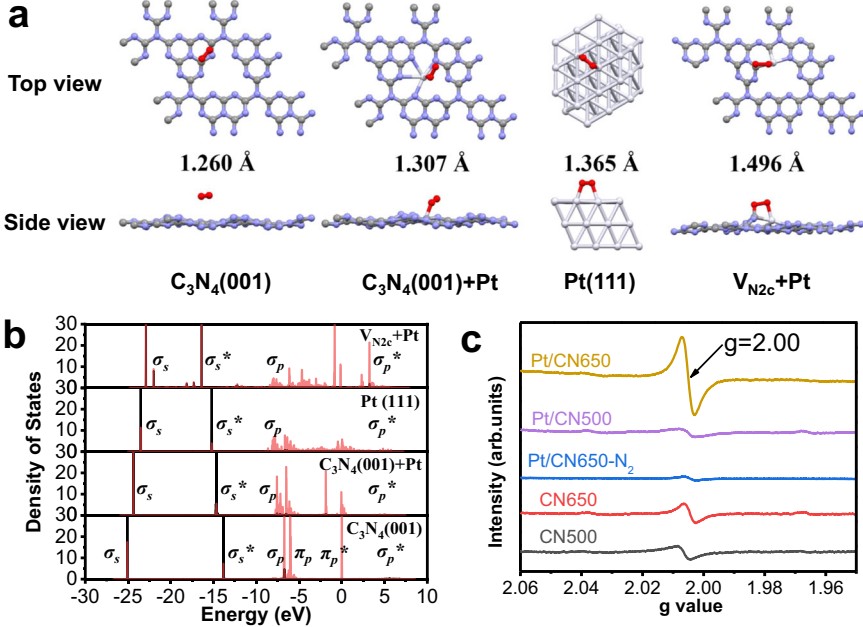

**Fig. 6 Mechanistic insights of catalytic toluene oxidation over the catalysts. a** The structures of $O_2$ adsorbed on pure $C_3N_4$ (001), $C_3N_4$ (001) +Pt, Pt (111), and $V_{N2c}$ + Pt (gray, purple, white, and red represents C, N, Pt, and O). **b** Partial density of states (PDOS) of the adsorbed $O_2$ (The black and red lines represent the O 2*s* and 2*p* orbitals, respectively. The molecular orbitals are also denoted). **c** EPR spectra of carbon nitrides and supported Pt catalysts at room temperature in air and $N_2$ atmosphere.

transfer is about 0.39, 0.21 electron or nearly zero, respectively. With electron further occupying the $\sigma_p^*$ state, the $\sigma_p$ bond of $O_2$ molecule is expected to break. The weak interaction of the bonding state ($\sigma_s$) and antibonding state ($\sigma_{s*}$) of the adsorbed $O_2$ on Pt + $V_{N2c}$ also suggests the elongation of the O–O bond, which facilitates the further reaction of O–O cleavage. Comparing the non-defective and defective cases, the contribution of nitrogen-vacancy should be the main reason to reduce the activation energy of the catalytic toluene oxidation.

In experiment, electron paramagnetic resonance (EPR) was carried out to investigate the capability of Pt/CN500 and Pt/CN650 for the activation of molecular oxygen. Exposed to air at room temperature, Pt/CN650 exhibit an obvious signal with a g factor of 2.00 (Fig. 6c). This signal can be ascribed to surface superoxide radical anion $O_2^{\cdot-}$ species. As for Pt/CN500, this signal is very weak. We also measure the EPR properties of Pt/CN650 under the nitrogen atmosphere. Only a weak signal can be observed in the absence of molecular oxygen. This result confirms that Pt/CN650 possesses excellent capability to activate molecular oxygen at low temperatures.

Very weak EPR signals were detected over the pure CN500 and CN650 supports. This means the unpaired electrons of carbon nitride support alone cannot effectively initiate the activation of molecular oxygen at low temperatures. Comparing the EPR signal for Pt/CN500 and Pt/CN650, the signal for Pt/CN500 is quite weak, although the Pt nanoparticles in the two samples are both in metallic states. This difference suggests that metallic Pt nanoparticles cannot independently undertake the responsibility of molecular oxygen activation at low temperatures either. Otherwise, the synergism of metallic Pt nanoparticles and nitrogen vacancies of carbon nitride support should be responsible for the high performance of $O_2$ activation at low temperatures.

## Discussion

In summary, nitrogen vacancies of carbon nitride significantly enhance the $O_2$ activation of Pt-based catalysts. The metallic Pt are well maintained over the manipulated carbon nitride. The Pt atoms and the dangling carbon atoms surround the vacancies synergistically donate electrons to the antibonding orbital of the adsorbed $O_2$, effectively rendering $O_2$ more reactive toward reactants and lowering the activation energy of the reaction. This work provides a facile manufacture way to fabricate highly efficient supported-Pt catalyst for catalytic oxidation. It should be noted that this strategy is of great potential to further enhance the activity. The small specific surface area of carbon nitride used in this work obviously limit the adsorption of toluene. Rapid progress in the synthesis of non-oxide materials would provide much opportunity. The adsorption and activation of pollutant molecules could be easily increased by tuning the texture properties of this non-oxide support. The concept of increased electron donors induced by nitrogen vacancy could be extended to develop a series of supported metal nanoparticles catalysts with diverse functionalities.

## Methods

**Materials.** Melamine and polyvinyl pyrrolidone (PVP) were purchased from Tianjin Guangfu Fine Chemical Research Institute. $H_2PtCl_6 \cdot 6H_2O$ was purchased from Sinopharm Chemical Reagent Co., Ltd. NaOH, HCl, and ethylene glycol were obtained from Beijing Chemical Works. All chemical reagents were analytical grade and without purification before used.

**Preparation of different carbon nitride supports.** Different carbon nitride samples were prepared by heat treatment of melamine precursors. Five gram of melamine was sealed in a ceramic crucible and placed into muffle furnace. It was heated up to corresponding temperatures (500, 525, 600, and 650 °C) with a heating rate of 10 °C min$^{-1}$, and kept at this temperature for 2 h. The product was

cooled down to room temperature and ground to powder. The obtained samples were denoted as CN500, CN525, CN600, and CN650, respectively.

**Preparation of Pt nanoparticles.** Pt nanoparticles were prepared by a polyol reduction method which was reported in our previous work. In a typical case, 15 mL NaOH-glycol solution (0.5 mol L$^{-1}$) was added into 27 mL $H_2PtCl_6 \cdot 6H_2O$-glycol solution (11.4 mmol L$^{-1}$). The pale-yellow transparent solution was heated at 90 °C for 2 h under $N_2$ atmosphere. After cooling to room temperature, HCl (0.77 mol L$^{-1}$) was dropwise added into the colloid until the pH value was adjusted to 3.0. The Pt nanoparticles were collected by centrifugation and subsequently dispersed in PVP-aqueous solution.

**Preparation of different Pt/CN catalysts.** The catalysts were prepared by an impregnation method. Typically, a certain amount of carbon nitride powder was added into the Pt PVP-aqueous solution and the mixture was stirred for 0.5 h, and subsequently evaporated to remove the water at 80 °C. The stoichiometrically content Pt was controlled to 0.3 wt%. The obtained samples were calcined at 250 °C in 20 vol%$O_2$/Ar flow for 2 h and the resultant powder was named as Pt/CN500, Pt/CN525, Pt/CN600, and Pt/CN650.

**Characterization.** X-ray diffraction (XRD) patterns were performed on an Empyrean X-ray diffractometer with Cu K$_\alpha$ radiation source. FT-IR measurements were operated on a Nicolet 6700 spectrometer. $N_2$-adsorption/desorption tests were carried out on the ASAP 2010N instrument. The surface areas were calculated based on the model of Brunauer–Emmett–Teller (BET). X-ray photoelectron spectroscopy (XPS) were recorded on a Thermo ESCA LAB 250 instrument. The high-resolution transmission electron microscopy (HRTEM) images were collected on a JEM-2100F instrument. Electron paramagnetic resonance (EPR) results were obtained on a JES-FA 200 EPR spectrometer. The UV–Vis diffuse reflectance spectra were measured on a Shimadzu-3600 UV–Vis–NIR spectrophotometer. The photoluminescence (PL) spectra were carried out on the FLS920 (Edinburgh Instrument) at room temperature and the excitation wavelengths were 380 and 365 nm. The emission decay lifetime was measured at 485 nm.

XAS measurements were performed using the Sector 9-BM beamline of the Advanced Photon Source at Argonne National Laboratory (Argonne, IL). The end station was equipped with a double-crystal Si (111) monochromator for wavelength selection. A 12-element Ge fluorescence detector was used to collect spectra. Sample powders were packed on plastic washers to enhance the signal. Data processing was performed using WinXAS software.

DRIFT spectrum was collected at 140 °C in 20 vol% $O_2$/Ar atmosphere. The sample was pretreated at 200 °C in $N_2$ for 15 min before testing. $C_7H_8$-TPD was obtained by subtracting the blank spectrum of the samples and the procedure were as follows. Firstly, 50 mg of the samples were pretreated at 200 °C in Ar flow for 30 min. After cooling to 70 °C, toluene was introduced for adsorption by gas bubbling method, the time for adsorption was 30 min. Then treating the samples were maintained at 70 °C in Ar flow for 1 h, and collected the signal with temperature rising. The blank spectrum of the sample is obtained in the same way except for the toluene adsorption.

## Method and models for the first-principles calculations

All the first-principles calculations in this work are performed by the VASP code[45,46]. The PAW pseudopotentials and optPBE-vdw functional are employed for structural optimization with an energy cutoff of 520 eV[47,48]. The $2 \times 2 \times 1$ slab models containing respective three atomic layers are adopted with the vacuum layer of 20 Å for $C_3N_4$ (001) and Pt (111). The bottom two layers and the lattice vectors are fixed, and the top one layer and the adsorbed atoms are fully relaxed with $2 \times 2 \times 1$ and $5 \times 5 \times 1$ Monkhorst-Pack k-point sampling until the force on each atom is less than 0.01 eV/Å for the $C_3N_4$ (001) and Pt (111) facet, respectively. Although complicated models with multiple N-vacancies and Pt atoms cannot be ruled out, simple cases including single type of defects and single Pt atom are found enough to describe the synergistic effect of the support and Pt in this work.

**Catalytic performance evaluation.** The catalytic activity tests were performed at a continuous-flow fix-bed reactor. For toluene oxidation, 100 mg of the catalyst with 40–60 mesh were placed into the central position of the quartz reactor. Toluene was introduced into the reactor by bubbling air stream through a toluene reservoir (keep in an ice bath). Another air stream was used to achieve toluene vapor concentration of 1000 ppm. The concentrations of reactants and products are analyzed by an online gas chromatograph equipped with an FID detector for toluene and a TCD detector for $CO_2$.

Toluene conversion ($X_{tol}$) is calculated as Eq. (1):

$$X_{tol} = \frac{C_{in} - C_{out}}{C_{in}} \times 100\% \tag{1}$$

Specific reaction rates for toluene oxidation at different temperatures are evaluated under the condition of toluene conversion is below 15%. For each run at a specific temperature, toluene conversions are averaged at the steady state and the specific

rates are calculated as Eq. (2):

$$r_{tol} = \frac{X_{tol} F_{tol}}{m_{cat}}, \tag{2}$$

where $X_{tol}$ is the toluene conversion at different temperatures, $F_{tol}$ is the molar flow rate of toluene and $m_{cat}$ is the mass of catalyst.

In the condition of excess oxygen, toluene oxidation has been reported following first-order kinetics to toluene concentration[49], which follows Eq. (3):

$$r = -kc \tag{3}$$

The apparent activation energies are calculated based on the Arrhenius Eq. (4):

$$\ln k = -\frac{E_a}{RT} + \ln A, \tag{4}$$

where $k$ is the rate constant, $E_a$ is the apparent activation energy, and $A$ is the pre-exponential factor.

As for CO oxidation, 80 mg of the catalyst with 40–60 mesh are used. The inlet reactant gas is 1 vol% CO, 5 vol% $O_2$, and Ar balanced. The total gas flow rate is 100 mL min$^{-1}$ and the space velocity is 75,000 mL g$^{-1}$ h$^{-1}$. The outlet reactants and products are analyzed by an online gas chromatograph equipped with a TCD detector. CO conversion ($X_{CO}$) is calculated as the formula (5):

$$X_{CO} = \frac{C_{in} - C_{out}}{C_{in}} \times 100\% \tag{5}$$

## Data availability

The data that support the findings of this study are available within the article and its Supplementary Information files. All other relevant data supporting the findings of this study are available from the corresponding authors upon request. Source data are provided with this paper.

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

## Acknowledgements

The authors acknowledge financial support from the National Science Foundation of China (22072054, 21972053, 11847043, and U1967215), the Development Project of Science and Technology of Jilin Province (20170101171JC and 20180201068SF), the 111 Project of China (B17020), the Open Project of State Key Laboratory of Inorganic Synthesis and Preparative Chemistry (202105) and the Fundamental Research Funds for the Central Universities. This research used resources of the Advanced Photon Source, an Office of Science User Facility operated for the US Department of Energy (DOE) Office of Science by Argonne National Laboratory, and was supported by the US DOE under contract no. DE-AC02-06CH11357, and the Canadian Light Source (CLS) and its funding partners. The authors also acknowledge the computational support from the Beijing Computational Science Research Center.

## Author contributions

T.G. and G.L. designed the experiment and performed the measurements and data analysis. D.M. and P.Z. contributed to the XAS measurements and discussion. W.Z. contributed to the catalytic tests and discussion. J.Y. and S.-H.W. contributed to the theoretical calculations. X.C., Y.Z., and W.Y. contributed to the characterizations. T.G., J.Y., and G.L. contributed to the discussion and the writing of the manuscript. All authors commented on the manuscript.

## Competing interests

The authors declare no competing interests.
