## [Peer Review File · Nature Communications]

Reviewers' Comments:

Reviewer #1:

Remarks to the Author:

The authors presented an interesting study of CN as a beneficial non-oxide support material for activating Pt catalyst for toluene oxidation. Using different temperature thermal treatment over the CN support, different N vacancies concentration in the support can be achieved, which were suggested to be responsible for the low temperature activation of toluene oxidation with the electron donor effect from the vacancy populated CN support after higher temperature (650oC) process. Overall, the structure and mechanistic analyses and discussion over the vacancy effect is rather sound and fluent. However, there are a few major issues. 1) in Figs. 1g and 1h, the absorbance vs. wavelength data was not exactly correlated with the PL data, where a band-to-band transition was interpreted; further, the decay of such 'band-edge' emission with increasing temperature of thermal treatment was explained to relate to the non-radiative pathway decay, should it be reversed, say increase of non-radiative pathway? 2) the excitation source wavelength of 380nm used for the PL seems to be too close to the band gap of CN materials? 3) the Pt dispersion trend was correlated with the CN treatment temperature as well as the N vacancies, however, is there preparation effect behind, for example, the PVP effect? 4) the durability results in Fig. 4 is not convincing. In Fig. 4b, the authors tried to demonstrate the good stability of the catalysts by presenting the 100% Toluene conversion of the fresh sample and the sample kept in the air for one year. However, leaving the catalysts in the air for one year does not necessarily indicate the good stability of the catalysts, proper references are needed to support the authors' statement if any. On the other hand, keeping the samples at 190C for 200 hours cannot demonstrate the excellent stability of oxidation catalysts, either. The practical VOC catalyst working conditions are much harsher than that. The catalytic performance of the catalysts before and after proper hydrothermal aging treatment would be helpful to demonstrate the stability of the catalysts. 5) in practice, S-poisoning could be another issue influencing VOC catalyst durability.

Reviewer #2:

Remarks to the Author:

The contribution of Gan et al. describes the development of defective carbon nitrides as support for Pt nanoparticles as a catalyst for oxygen activation.

It is indeed true that carbon-nitrides, N-doped carbons, Covalent Triazine Frameworks and other related materials have been gaining attention as potential alternative catalytic materials. Mostly the focus is also on the aspect of the "metal-free" catalysis. Metal-free oxygen activation on a CTF has been recently demonstrated (10.1126/sciadv.aaz2310) and it has been earlier reported for N-doped carbons (10.1021/acscatal.5b00375). Antonietti and his team coined the active sites of such catalysts as "noble carbons" (10.1002/adma.201706836), and proved that these sites were able to oxidize gold. The these noble sites were identified as the "graphitic carbon", bearing a positive charge on the N. Strangely, this word "graphitic carbon", occurs 9 times in the manuscript, 9 times in the titles of the references, but not once in the actual manuscript.

It would be interesting to see how the C₃N₄ materials without the metal sites would perform. Authors have done some EPR measurements, arguing that there is a synergy between both. But based upon all these and more recent reports, one could hope that the metal free variant would also work.

The manuscript is quite long and reads much more than a full paper than as a communication. While it is a very good manuscript, I do not see the urgency of this publication. All new concepts have already been reported. It also lacks that brilliant new insight you would expect from a Nature Communication. I would suggest the authors rewrite the work as a full paper, hopefully with the above suggestions in mind, and submit this to a (good!) catalysis or materials journal.

Finally, I express my appreciation to the authors for the thorough work embodied in this paper.

Reviewer #3:

Remarks to the Author:

The authors study Pt particles supported on N-defective C₃N₄ as an oxidation catalyst. The catalyst is characterized using a variety of experimental techniques and shown to be highly active toward toluene oxidation at temperatures below 200 °C. The authors also report high catalyst stability. These findings are supported by DFT calculations for a model catalyst.

The experimental work is thorough, and the reported catalyst shows promising oxidation activity. However, the description of the computational work requires more details, and the connection between computational and experimental work is weak. The manuscript might be suitable for publication once the following comments have been addressed.

(1) The experimental data clearly shows larger particles, whereas the DFT work studies single atoms only. Given that the properties of metal clusters vary substantially with cluster size, I am not convinced that this is a suitable model system.

(2) On page 5, the authors use DFT energies at 0 K to justify observations at elevated temperatures. This should be worded more carefully given that entropies are not considered.

(3) The top views in Fig 6a are rather convoluted. It would help if the authors only showed the topmost layer plus the O₂ molecule.

(4) Pt atoms can bind to non-defective C₃N₄, which is amply discussed in the literature. When discussing the impact of the N defects, comparing to the defect-free site is an important step that would elevate the present manuscript.

(5) Also, did the authors consider a N₃c structure where the Pt atom binds in the larger N₆ cavity?

(6) The authors need to comment on the convergence of their energies regarding parameters such as the the number of k points. Right now, it appears like their calculations for Pt(111) do not use enough k points.

Response to Reviewers

Reviewer #1:

The authors presented an interesting study of CN as a beneficial non-oxide support material for activating Pt catalyst for toluene oxidation. Using different temperature thermal treatment over the CN support, different N vacancies concentration in the support can be achieved, which were suggested to be responsible for the low temperature activation of toluene oxidation with the electron donor effect from the vacancy populated CN support after higher temperature (650 °C) process. Overall, the structure and mechanistic analyses and discussion over the vacancy effect is rather sound and fluent. However, there are a few major issues.

1) in Figs. 1g and 1h, the absorbance vs. wavelength data was not exactly correlated with the PL data, where a band-to-band transition was interpreted; further, the decay of such 'band-edge' emission with increasing temperature of thermal treatment was explained to relate to the non-radiative pathway decay, should it be reversed, say increase of non-radiative pathway?

Response: We agree with the reviewer on these two points and correct them in the revision. We realize that the labeling of peak center in our previous manuscript might mislead the meaning. The peak center did not reflect the band-to-band transition. Indeed, the band-to-band transition shift in such wide range is impossible. In the revision, the labeling of peak center was removed (see Figure 1h, Supplementary Figure 3, and Supplementary Figure 15). Instead, a dash line was used to compare the main signals.

In addition, we revised confusing descriptions on PL spectra. Using “Increase of non-radiative pathway” is correct (see page 7 line 15 to 18). Thanks for the helpful correction.

2) the excitation source wavelength of 380 nm used for the PL seems to be too close to the band gap of CN materials?

Response: We make complement for PL measurement in the revision, but first we would like to give some explanation on the test conditions used in previous version. The excitation source wavelength was determined according to the light-absorption properties of CN materials. We double checked the UV-vis spectra of all these four samples. In principle, wavelength of 380 nm could meet the requirement of PL measurements for our samples. We understand the reviewer's concern about the potential influence of the excitation source on the analysis of final spectra. In the revision, PL spectra of both CN and Pt/CN samples are further measured with excitation source wavelength of 365 nm. Similar results were obtained for both CN and Pt/CN samples (see Figure R1 and Figure R2). So, we hope to retain original results in the main text and add these complement results in the supplementary information. (see Supplementary Figure 3 and Supplementary Figure 15b).

Figure R1. Photoluminescence (PL) emission spectra, excitation wavelength: 365nm.

Figure R2. Photoluminescence (PL) emission spectra of Pt supported on different carbon nitriles, excitation wavelength: 365nm.

3) the Pt dispersion trend was correlated with the CN treatment temperature as well as the N vacancies, however, is there preparation effect behind, for example, the PVP effect?

Response: In the process of catalyst design, we had fully considered the factors that might affect the discussion on the intrinsic properties of Pt/CN catalysts, especially the condition to load Pt on the CN support. As the reviewer concerned, this process is a critical step for understanding the CN support effect. The ideal condition is to maintain the original surface properties of different CN materials. We chose a colloid deposition method to load the size-controlled Pt nanoparticles. On one hand, it could exclude the size effect of Pt nanoparticles. On the other hand, this method could avoid the *in-situ* generation of Pt nanoparticles changed the surface properties of CN. A series of characterizations demonstrated that the surface properties of Pt/CN still maintain the change trend of CN supports (Supplementary Figure 6, 8-16, Supplementary Table 5 and 6).

The role of PVP is to stabilize the Pt nanoparticles in H₂O. In the presence of PVP, the homogeneity of Pt colloid could be maintained for several months. Without PVP, Pt nanoparticles would deposit after one day (see Figure R3). PVP could facilitate to the dispersion of Pt nanoparticles on the surface of CN supports. From this aspect, PVP has

positive effect on the preparation of Pt/CN. However, PVP could not intrinsically affect the catalytic performance of Pt/CN. For comparison, we prepared Pt/CN650 and Pt/CN500 with Pt colloid in the absence of PVP. Similar change trend is observed both in toluene oxidation over these two samples (Figure R4 or Supplementary Figure 25). It showed that PVP effect would not affect the analysis on the role of CN support in enhancing the activity. We added these results and corresponding description in the revision (see page 15 line 2 to line 4).

Figure R3. Pt nanoparticles colloid prepared with and without PVP.

Figure R4. Toluene conversion as a function of temperature over two typical samples.

4) The durability results in Fig. 4 is not convincing. In Fig. 4b, the authors tried to demonstrate the good stability of the catalysts by presenting the 100% Toluene conversion of the fresh sample and the sample kept in the air for one year. However, leaving the catalysts in the air for one year does not necessarily indicate the good

stability of the catalysts, proper references are needed to support the authors' statement if any. On the other hand, keeping the samples at 190°C for 200 hours cannot demonstrate the excellent stability of oxidation catalysts, either. The practical VOC catalyst working conditions are much harsher than that. The catalytic performance of the catalysts before and after proper hydrothermal aging treatment would be helpful to demonstrate the stability of the catalysts.

Response: We agree with the reviewer that hydrothermal aging would be a much harsher condition to investigate the stability of the catalysts. In the revision, the catalytic tests before and after hydrothermal aging treatment are added. As shown in Figure R5, we firstly detect the activity of the Pt/CN650 catalyst at 190°C for 5 h. Subsequently, we increase the temperature to 250°C and introduce 20 vol% H₂O into the reactor. The catalyst is treated under this condition for 6 h. Finally, the temperature is returned to 190°C to investigate the catalytic performance. The result shows that the Pt/CN650 catalyst represented almost the same activity as that before treatment. We repeated this experiment. The difference is that we added 30 vol% H₂O into the reactor. The catalyst still represents the activity like fresh one.

Figure R5. Stability test of Pt/CN650 after different hydrothermal treatments.

In addition, we would like to explain the meaning of durability results in our work. We investigated the catalytic performance of the catalyst after a long-term storage

mainly according to the feedback from our partners in emission-control industry. They hoped the catalysts could meet the requirement of ready-to-use, that is, the storage catalysts can be put into use without any pretreatment. Besides, most of emission-control equipment is used intermittently. The catalyst would be exposed to air for a long time before next run. This point is a little different from the catalysts used in chemical industry. We also noticed some work had carried out similar test, for example, Chen's report on CO oxidation (*Science* 2014, 344, 495). So, we hope to retain this result in the revision.

In comparison, the reviewer suggested a much harsher condition to test the durability of catalysts, which could obviously shorten the test time. We add these results and corresponding analysis in the revision (see page 11 line 7 to page 12 line 2). Hydrothermal aging test could further elevate the quality of this work. Thank the reviewer for the helpful suggestion. We will also consider this test in our following work.

5) in practice, S-poisoning could be another issue influencing VOC catalyst durability.

Response: We agree with the reviewer that the influence of S-poisoning is an important issue in many practical applications. It could significantly affect the performance of metal and metal oxide catalysts. In our case, the influence of S-poisoning on the catalytic performance is investigated with gas mixture containing 50 ppm SO₂. The reaction was carried out at 190°C and 210°C, respectively. As shown in Figure R6, the introduction of SO₂ could decrease the activity of Pt/CN. The impact could be relieved when the reaction temperature increase to 210°C. When switch off the SO₂ flow, the conversion of toluene recover to 100 % immediately at 210°C. While at 190°C, the recovery time is relatively long. These results indicated that high reaction temperature could reduce the impact of SO₂.

Above results show that non-oxide supports cannot change the intrinsic properties of Pt sites. The CN support mainly contributed to the activation of O₂. Nevertheless, Pt-

based catalysts still has wide space for application in emission-control area. Our group had ever developed a Pt-based catalyst (not this work) for a furniture factory for VOCs control (sulfur-free). The equipment had run more than two years and still could meet the discharge standard of air pollutants. Based on S-poisoning results, a desulfurization device should be considered when the catalysts were used in a S-containing environment.

Figure R6. Stability test of toluene oxidation over Pt/CN650 in the presence of SO₂.

Reviewer #2:

The contribution of Gan et al. describes the development of defective carbon nitrides as support for Pt nanoparticles as a catalyst for oxygen activation.

It is indeed true that carbon-nitrides, N-doped carbons, Covalent Triazine Frameworks and other related materials have been gaining attention as potential alternative catalytic materials. Mostly the focus is also on the aspect of the "metal-free" catalysis. Metal-free oxygen activation on a CTF has been recently demonstrated (10.1126/sciadv.aaz2310) and it has been earlier reported for N-doped carbons (10.1021/acscatal.5b00375). Antonietti and his team coined the active sites of such catalysts as "noble carbons" (10.1002/adma.201706836), and proved that these sites were able to oxidize gold. The these noble sites were identified as the "graphitic carbon", bearing a positive charge on the N. Strangly, this word "graphitic carbon", occurs 9 times in the manuscript, 9 times in the titles of the references, but not once in the actual

manuscript.

It would be interesting to see how the C₃N₄ materials without the metal sites would perform. Authors have done some EPR measurements, arguing that there is a synergy between both. But based upon all these and more recent reports, one could hope that the metal free variant would also work.

The manuscript is quite long and reads much more than a full paper than as a communication. While it is a very good manuscript, I do not see the urgency of this publication. All new concepts have already been reported. It also lacks that brilliant new insight you would expect from a Nature Communication. I would suggest the authors rewrite the work as a full paper, hopefully with the above suggestions in mind, and submit this to a (good!) catalysis or materials journal.

Finally, I express my appreciation to the authors for the thorough work embodied in this paper.

Response: We carefully considered the reviewer's comments, in which we could feel the reviewer's positive evaluation to the quality of this work. We agree well with the reviewer's analysis around N-containing carbon materials. In the past decade, it is a constant hot topic in material and catalysis science. Some N-containing carbon materials have been developed as metal-free catalysts. We understand that it is a valuable subject in heterogeneous catalysis. The difference is the starting-point of this work, which mainly focused on the development of efficient Pt-based catalysts for emission-control. This subject was established in our lab in 2008, aiming to solve the problems in industrial application. As one of typical volatile organic compounds (VOCs) pollutants, gaseous state toluene is not only harmful to human health, but also causes environment problems such as photochemical smog and ozone pollution. Undoubtedly, Pt-catalyst is one of most effective catalysts for emission-control. The challenge is mainly the cost (determined by Pt contents), low-temperature activity and durability. Nearly all researchers expect to achieve complete conversion of pollutants at the lowest possible reaction temperature with lowest amount of Pt loading.

In our recent work, through finely tuning the mentality of Pt sites and surface acid-base properties of Al₂O₃ supports, we had reported a desirable Pt/Al₂O₃ catalyst for toluene oxidation (*Appl. Catal. B: Environ.* 2019, 257, 117943). The activity of Pt/Al₂O₃ (with 0.1wt%Pt) could be optimized to a leading position among oxide-supported catalysts. To be honest, we did not expect Pt/CN (carbon nitride-supported Pt catalyst, this work) could exceed Pt/Al₂O₃ when we designed this set of catalyst. One reason is that we had sieved a great deal of supports (mainly oxides) when we optimized the Pt/Al₂O₃. Another is that most of carbon materials has natural disadvantage in catalytic oxidation. At a certain temperature, conventional carbon would be oxidized during the reaction. However, we believe that the breakthrough must establish on the new ideas and/or materials. Therefore, non-oxides support were considered to be utilized for developing next generation Pt-based catalysts.

Indeed, high stability was only a small surprise from Pt/CN. The striking advance of Pt/CN was the decrease of activation energy from 65.9 kJ•mol⁻¹ (optimized Pt/Al₂O₃, Figure R7) to 46.3 kJ•mol⁻¹ (Pt/CN, in this work, Pt contents have little effect on the activation energy, Figure R8). That was an unexpected result, which essentially changed the thermodynamic properties of Pt-catalyst. It is well-known that the change of thermodynamic is difficult in catalysis. Sufficient evidence demonstrated that this change come from the contribution of the optimized CN support, relying on electron donor enhancement induced by nitrogen vacancy. This point was quite different from conventional oxide support. It can be recognized as a transformative advance for both the development of Pt-based catalysts and application of CN materials. Obviously, this strategy could be extended to design other active oxidation catalysts. Due to the high impact of *Nature Communications* as well as the open access to the readers, we hope this strategy could be delivered from this journal to more researchers.

Figure R7. Arrhenius plots for toluene oxidation over different Pt/Al₂O₃ catalysts. (*Appl. Catal. B: Environ.* 2019, 257, 117943)

Figure R8. Arrhenius plots for toluene oxidation over Pt/CN650 with different loading amount of Pt. (*this work*)

In addition, we would like to talk some thinking and experience on the development of metal-free and non-noble metal oxidation catalysts. We might have the same interest on this valuable topic as the reviewer. To our knowledge, under similar reaction condition, the activity and durability of non-noble metal catalysts still much lower than noble metal catalysts in most of VOCs control reactions. Such kind of catalysts are also investigated in our lab. In practical applications, we need to balance the importance between the cost of noble metal catalysts and energy efficiency of non-noble metal

catalysts. After all, the energy consumption cannot be ignored in the total running cost.

During we prepared this work, the catalytic performance of CN materials, as metal-free samples, was also investigated. Only 6% conversion of toluene could be observed at reaction temperature of 300°C (Figure R9), which is much lower in comparison with Pt/CN. Similar result was also observed in CO oxidation (Figure R10). We noticed that some N-containing carbon materials could catalyze some oxidation reactions at low temperature. It should be noted that most of these reports focused on the oxidation of different alcohols. In our previous work, based on understanding O₂ activation mechanism over different materials, we had carried out some similar reaction over a few carbon materials (metal-free, *J. Colloid Inter. Sci.*, 2014, 421, 71; *J. Colloid Interface Sci.* 2010, 342, 467) and metal oxides catalysts (*J. Catal.*, 2019, 377, 145; *Chem. Commun.*, 2016, 52, 13495). In comparison with catalytic toluene oxidation, alcohol oxidation is relatively easy even under mild conditions. β-hydrogen atom from the alcohol could be abstracted by active $\bullet\text{O}_2$ forming aldehydes or ketones. We had thought about the possibility of metal-free catalysts in toluene oxidation. Maybe photo- or electro- driven system could make up the lack of sufficient capacity of metal-free samples. But under current thermo-driven system, the synergism between metal sites and supports have to be considered.

Figure R9. Toluene conversion as a function of temperature over carbon nitrides (metal free).

Figure R10. CO conversion as a function of temperature over carbon nitrides (metal free).

As for not using "graphitic carbon" to name our samples, we had our own consideration when we prepared this article. The main reason was that not all these four samples possessed typical structure of graphitic carbon (see Figure 1b). "Carbon nitride" is more suitable for denoting our samples. Besides, the nitrogen-vacancies result in the increase of electron donors, which is different from perfect "noble carbon". We think that the combination of active surface properties and intrinsic structural stability endow the CN material as excellent support for Pt nanoparticle.

We realize that the manuscript is long in comparison with a conventional communication. In the revision, we reduce the length of the manuscript as possible as we can. We estimated that it could be put in 8 pages in the final version. We double checked the recent article published in *Nature Communications*. The length of the revision could meet the requirement of publication.

In summary, this work brings a transformative advance for both the development of Pt-based catalysts and application of CN materials. The novelty includes not only the change of thermodynamic properties of Pt-catalyst, but also extraordinary stability after long-term storage and under harsh conditions (see Figure R5, response to question 4 of Reviewer#1). As suggested by the other two reviewers, our paper has the broad interest and importance. It would stimulate the researchers re-thinking the design of supported noble metal catalysts. We hope this idea could be delivered from *Nature*

Communications to researchers.

Reviewer #3:

The authors study Pt particles supported on N-defective C₃N₄ as an oxidation catalyst. The catalyst is characterized using a variety of experimental techniques and shown to be highly active toward toluene oxidation at temperatures below 200 °C. The authors also report high catalyst stability. These findings are supported by DFT calculations for a model catalyst.

The experimental work is thorough, and the reported catalyst shows promising oxidation activity. However, the description of the computational work requires more details, and the connection between computational and experimental work is weak. The manuscript might be suitable for publication once the following comments have been addressed.

Response: Thanks for the appreciation that our experiment work is thorough and promising. We realize the description of computation and its correlation to the experiment is insufficient due to the word limit. Therefore, in this revision, we have revised several parts to address the correlation.

(1) The experimental data clearly shows larger particles, whereas the DFT work studies single atoms only. Given that the properties of metal clusters vary substantially with cluster size, I am not convinced that this is a suitable model system.

Response: We understand the consideration of the reviewer on the computational model and we agree that the size of the Pt cluster may affect the activation of O₂. However, our work focused on the electron donation of N-vacancies. In our experiments, sufficient evidence had demonstrated the critical role of CN with N-vacancies instead of Pt content in boosting O₂ activation. This simple model in the manuscript is established to be a prototype model to discuss the synergism between N-

vacancy and Pt. And as shown in the updated Figure 6 (or Figure R11), comparing the O₂ activation on different catalysts with and without N-vacancy (suggested by the reviewer in question 4), it is clearly disclosed that Pt/CN with N-vacancy is beneficial to donate electron to the O₂ and thus facilitates the whole reaction. The simple model in our manuscript is enough to discuss the synergistic effect. On the other hand, if comparing the O₂ adsorption on Pt (111) and C₃N₄(001)+Pt in Figure 6, the activation of O₂ on Pt (111) is clearly preferred. Therefore, it is expected that the activation of O₂ would be possibly enhanced with Pt clusters combined more N-vacancies. Limited by our computational resources, searching and confirming the uncertain real atomic structure is meaningful but far from our original intention. Therefore, we use a simple model to indicate the detailed atomic and electronic structure of activated O₂ on different catalytic surface, especially the one on the V_{N2c}+Pt.

One sentence “Although complicated models with multiple N-vacancies and Pt atoms cannot be ruled out, simple cases including single type of defects and single Pt atom are found enough to describe the synergistic effect of the support and Pt in this work.” is added in methods section (Supplementary page 6 line 16 to 19).

(2) On page 5, the authors use DFT energies at 0 K to justify observations at elevated temperatures. This should be worded more carefully given that entropies are not considered.

Response: We agree that entropies of gas should be carefully considered. In this manuscript, the key step of the reaction is the adsorption and activation of O₂ on the catalyst, which is amply studied in previous literature (*Chem. Rev.* 2018, 118, 2816). Before and after the adsorption, the entropy change on different catalysts is dominantly determined by the entropy of O₂, because no phase transition is found except for O₂ (from gas to solid). Comparing the activated O₂ on different catalysts in this work, the entropy difference of the O₂-adsorped catalytic surfaces could be neglected due to the intrinsic small entropy of solid-state matter.

(3) The top views in Fig 6a are rather convoluted. It would help if the authors only showed the topmost layer plus the O₂ molecule.

Response: Thanks for the advice, we have modified the figure as the reviewer suggested (see Figure 6 in the revision).

Figure R11. Mechanistic insights of catalytic toluene oxidation over the catalysts.

a The structures of O₂ adsorbed on pure C₃N₄ (001), C₃N₄ (001)+Pt, Pt (111), and V_{N_{2c}}+Pt (grey, purple, white, and red represents C, N, Pt, and O). **b** Partial density of states (PDOS) of the adsorbed O₂ (The black and red lines represents the O 2s and 2p orbitals, respectively). The molecular orbitals are also denoted). **c** EPR spectra of carbon nitriles and supported Pt catalysts at room temperature in air and N₂ atmosphere.

(4) Pt atoms can bind to non-defective C₃N₄, which is amply discussed in the literature. When discussing the impact of the N defects, comparing to the defect-free site is an important step that would elevate the present manuscript.

Response: It is a helpful complement to discuss the impact of the N-vacancies. In the revision, we considered several cases of the catalyst (including the case of Pt atom bond to the pure C₃N₄ surface suggested by the referee) and investigated the O₂ adsorption on the corresponding surface (see Figure R11). The results show that the adsorption of O₂ on the vacancy-free Pt/CN is only slightly enhanced comparing to the direct adsorption on the pure CN, but much weaker than that on the surface of vacancy-containing Pt/CN. This result further confirms the critical role of N-vacancy in the activation of O₂. Corresponding modification was added in the revision (see page 15- page 16 highlighted). Thanks for the helpful suggestion.

(5) Also, did the authors consider a N_{3c} structure where the PT atom binds in the larger N₆ cavity?

Response: We had considered several models of V_{N_{3c}}+Pt including the one referred. The result shows that Pt atom preferred to bind to the vacancy of N_{3c} rather than the N₆ cavity by 0.25 eV.

Figure R12. Optimized structures of the V_{N_{3c}}+Pt

(6) The authors need to comment on the convergence of their energies regarding parameters such as the the number of k points. Right now, it appears like their calculations for Pt (111) do not use enough k points.

Response: We are sorry that we failed to mention the K-points for Pt (111) and C₃N₄ (001) in the previous manuscript. The Pt(111) and C₃N₄(001) are optimized by with 5×5×1 and 2×2×1 Monkhorst-Pack k-point sampling, respectively. In the revision, we have modified the sentence into “The bottom two layers and the lattice vectors are fixed, and the top one layer and the adsorbed atoms are fully relaxed with 2×2×1 and 5×5×1 Monkhorst-Pack k-point sampling until the force on each atom is less than 0.01 eV/Å for the C₃N₄ (001) and Pt (111) facet, respectively.”(see page 6 line 13 to line 16 in supplementary information)

Reviewers' Comments:

Reviewer #1:

Remarks to the Author:

The authors have reasonably addressed the reviewers' concerns raised in the revision through conducting a series of additional experiments and analyses, however there are some grammar and spelling issues throughout the manuscript, which should be corrected before acceptance for publishing.

Reviewer #2:

Remarks to the Author:

The authors have written a very convincing rebuttal, adding many details and explanations. I was truly impressed by the depth of this rebuttal letter. I can say that all my concerns have been addressed satisfactorily. The manuscript is modified adequately.

Reviewer #3:

Remarks to the Author:

The authors have addressed my comments adequately. I therefore recommend publishing the article in Nature Communications.

Response to Reviewers

Reviewer #1 (Remarks to the Author):

The authors have reasonably addressed the reviewers' concerns raised in the revision through conducting a series of additional experiments and analyses, however there are some grammar and spelling issues throughout the manuscript, which should be corrected before acceptance for publishing.

Response: We have polished the language and corrected the grammar errors in the revision. Thank you for your recommendation.

Reviewer #2 (Remarks to the Author):

The authors have written a very convincing rebuttal, adding many details and explanations. I was truly impressed by the depth of this rebuttal letter. I can say that all my concerns have been addressed satisfactorily. The manuscript is modified adequately.

Response: Thank you for your recommendation.

Reviewer #3 (Remarks to the Author):

The authors have addressed my comments adequately. I therefore recommend publishing the article in Nature Communications.

Response: Thank you for your recommendation.